# Exposure-Based Intervention in Virtual Reality to Address Kinesiophobia in Parkinson’s Disease: A Narrative Review

**DOI:** 10.3390/jcm14248837

**Published:** 2025-12-13

**Authors:** Alice Jeanningros, Stéphane Bouchard, Alexandra Potvin-Desrochers

**Affiliations:** 1Psychoeducation and Psychology Department, Université du Québec en Outaouais (UQO), Gatineau, QC J8X 3X7, Canada; alicejeanningros@gmail.com (A.J.); stephane.bouchard@uqo.ca (S.B.); 2Psychosocial Medicine Research Center, Centre Intégré de Santé et Services Sociaux de l’Outaouais (CISSSO), Gatineau, QC J8T 4J3, Canada; 3Interdisciplinary School of Health, Université du Québec en Outaouais (UQO), 283 Boul. Alexandre-Taché, Gatineau, QC J8X 3X7, Canada

**Keywords:** kinesiophobia, exposure-based intervention, virtual reality, Parkinson’s disease

## Abstract

**Background/Objectives**: Physical activity alleviates symptoms and may slow Parkinson’s disease (PD) progression, yet many individuals with PD remain sedentary. Kinesiophobia, the fear of movement, may represent a significant but underexplored psychological barrier to physical activity in this population. Virtual reality (VR), already effective in phobias, may represent a promising approach to address this challenge. This review initially aimed to systematically examine exposure-based interventions in VR (E-IVR) directly targeting kinesiophobia in PD. **Methods**: Database searches using keywords such as “kinesiophobia,” “fear of movement,” and “VR” combined with “PD” yielded no eligible studies. Consequently, the scope was broadened to include populations with neurological or musculoskeletal conditions, and a narrative review format was adopted to synthesize the available evidence. Furthermore, relevant studies of interventions in VR applied in PD, although not specifically addressing kinesiophobia, are detailed to provide evidence of efficacy and feasibility of VR interventions in PD. Finally, directions are offered to support the creation of E-IVR targeting kinesiophobia in individuals with PD. **Results**: Meta-analyses in neurological and musculoskeletal populations demonstrate moderate to large reductions in kinesiophobia following VR interventions, although effects vary depending on assessment tools, degree of immersion, and exposure design. In PD, VR has been applied to rehabilitation, anxiety reduction, and quality of life enhancement. These interventions achieved high adherence (≥90%), were well tolerated, and reported no major adverse events. **Conclusions**: Kinesiophobia is prevalent in PD and could contribute to physical inactivity. E-IVR appears feasible, safe, and innovative for addressing kinesiophobia in people living with PD.

## 1. Introduction

Parkinson’s disease (PD) is characterized as a chronic neurodegenerative disorder with a global impact growing significantly over the past decades. In 2019, more than 8.5 million cases have been reported worldwide, and the disease has caused 329,000 deaths representing an increase of over 100% since 2000 [1]. PD is caused by a progressive death of dopaminergic neurons leading to a loss of dopamine [2]. The latter has been demonstrated to induce alterations in the associated dopaminergic pathways, consequently culminating in the manifestation of motor and non-motor symptoms. Motor symptoms, such as bradykinesia, tremor, postural instability, and muscle rigidity, and non-motor symptoms, including psychiatric, gastrointestinal, urinary, and sleep-related disturbances, altogether characterize PD. Individuals living with PD have a significantly higher risk of falling than healthy age-matched peers, along with reduced quality of life and psychosocial well-being [2,3].

Currently, no cure exists for PD. Instead, the symptoms are managed through pharmacotherapy with the administration of levodopa being the gold standard. While it can be beneficial for several symptoms, the effectiveness of pharmacotherapy can vary across different disease subtypes and side effects can occur, including dyskinesias and impulse control disorders. Furthermore, there is an absence of high-quality evidence for many non-motor symptoms [2]. As a result, non-pharmacological interventions such as exercise, physical therapy, occupational therapy and speech therapy, have brought increased attention and have been demonstrated to be efficacious in the management of symptoms. Among these, physical activity is no longer considered merely promising but has been found to be effective in attenuating motor symptoms as confirmed with several recent systematic reviews and meta-analyses [4,5]. More precisely, home-based aerobic exercise has been shown to significantly reduce off-state motor signs in individuals with mild PD [6]. A high-intensity treadmill exercise in patients with de novo PD was found to be feasible, safe and led to significantly less worsening of motor symptoms over six months compared to the usual care group [7]. A recent meta-analysis (2023) further confirmed the positive overall impact of exercise on PD symptoms, with similar effectiveness between aerobic and non-aerobic exercise interventions (effect size d-index 0.155, Confident Interval ((CI) 95% of 0.0791 < µ < 0.2308) [8]. Such interventions not only alleviate motor symptoms but also enhance quality of life [9,10] and contribute to improvements in non-motor symptoms, with aerobic exercise producing significantly greater effect than conventional non-aerobic exercises [11]. Finally, initiating physical activity post-diagnosis also appears to lower the risk of all-cause mortality in PD compared to continued inactivity [12]. Together, these findings underscore the importance of physical activity as a therapeutic intervention with potential disease-modifying effects on PD.

Despite the positive impact that physical activity has on PD, a large proportion of patients remains sedentary. Studies have estimated that more than half of people living with PD are physically inactive, and among those who are active, 29% engage in less physical activity compared to healthy controls [13,14]. Low levels of physical activity have been associated with common barriers to exercise in PD, which span multiple domains including physical discomfort during exercise, fatigue, apathy, low prior activity levels or difficulty integrating exercise into daily routines, personal factors like low self-efficacy, fear of falling, and low outcome expectations, as well as environmental barriers such as limited social support and poor accessibility [15,16].

Kinesiophobia, an irrational and debilitating fear of physical movement [17], may be an under studied barrier to physical activity in PD. Up to 77.3% of PD patients may experience some level of kinesiophobia, which is significantly higher than in age-matched individuals without PD [18]. The concept of kinesiophobia was originally developed in the context of chronic low back pain [19], where fear is typically rooted in previous painful experiences, catastrophic thinking, and negative beliefs about movement. However, its clinical presentation may differ in PD, as the fear of movement is not necessarily triggered by pain and yet be perceived as rational and justified by patients due to their history of falls and balance impairments. In PD, kinesiophobia negatively correlates with levels of physical exercise and balance [20], and positively correlates with fear of falling, symptoms of anxiety and depression, severity of motor symptoms and age [18,21,22]. Incidence of kinesiophobia also seems to increase through progression disease stages [21]. While the exact origins of kinesiophobia in PD remain unclear, several potential contributing factors can be inferred from the literature, such as gait impairments and risk of falls [18,20] and a reduced belief in one’s ability to perform movements effectively [23]. In the PD literature, clinicians and researchers may also refer kinesiophobia to related constructs such as “fear of movement,” or “movement avoidance.” In this review, we consistently employ the term “kinesiophobia” when introducing and discussing the concept ourselves. However, alternative terms may appear when reflecting the original wording used by authors of the discussed articles. In such cases, they are reported faithfully but considered to fall under the broader conceptual umbrella of kinesiophobia.

While physical activity is essential for managing PD, with potential benefits for slowing disease progression [8], the psychological barrier of kinesiophobia remains largely unaddressed in interventions aimed at promoting physical activity. Combining psychological strategies targeting kinesiophobia with the proven benefits of virtual reality (VR) may represent an effective approach to increasing physical activity in individuals with PD. Indeed, VR has shown promising results across various domains, with evidence supporting its positive impact on medical training, therapeutic interventions, and patient care, including rehabilitation [24,25,26,27,28]. A review by Thangavelu et al. [29] identified 32 studies on VR use in PD; however, none directly assessed or targeted kinesiophobia. Thus, the purpose of this review was two folded: first, to summarize the existing literature on interventions in VR designed to reduce kinesiophobia in individuals with neurological conditions, such as chronic pain or PD; and second, to explore the potential of VR to enhance psychological interventions based on exposure specifically targeting kinesiophobia in PD. Leveraging exposure-based interventions in VR (E-IVR) to address the psychological mechanisms of kinesiophobia may provide a promising, non-pharmacological strategy to support physical activity engagement in this population.

This literature review was conducted using relevant keywords, including “kinesiophobia,” “fear of movement,” “fear of reinjury,” and “motion phobia,” in combination with terms such as “VR”, “virtual medicine”, “augmented reality”, “mixed reality”, and “reality virtuality” and “PD”. These keywords were used to conduct targeted searches in four major databases, MEDLINE, Scopus, PsycArticles, and SPORTDiscus, from December 2024 to April 2025. The initial search specifically targeted studies involving individuals living with PD. However, as this yielded no results, the scope was broadened to include studies conducted in other populations with neurological or musculoskeletal conditions. The expectation to conduct a systematic review was also revised to report a narrative review, which is better suited for an under investigated domain. Search strategies were progressively adapted using five complementary query combinations: (1) kinesiophobia and PD and VR; (2) kinesiophobia and PD; (3) kinesiophobia and VR; (4) VR and PD. In parallel, given the conceptual proximity between kinesiophobia and fear of falling in PD, an additional search targeting fear of falling, VR, and PD was performed. Together, these five complementary search strategies allowed us to identify relevant evidence on VR and movement-related fear in neurological and musculoskeletal populations when PD-specific data were unavailable, while maintaining conceptual alignment with the objectives of this narrative review. A systematic approach was followed to identify, select and screen articles.

After removing duplicates, a total of 1586 articles were identified. After title-level screening, 41 articles were retained for full-text review. Our primary objective was to identify studies combining VR and kinesiophobia; therefore, these five studies were screened and analyzed first. All VR and kinesiophobia studies that were not part of meta-analytic datasets were subsequently examined in detail and discussed in the narrative review section, whereas the meta-analyses themselves are reported globally rather than through individual study descriptions. Once this primary objective was addressed, the remaining full-text articles, predominantly involving VR and PD, were reviewed and further filtered to retain only those addressing psychological aspects or interventions relevant to movement-related fear or kinesiophobia.

This narrative review is structured as follows. First, the concept of kinesiophobia is introduced, with emphasis on its limited investigation and lack of a precise definition in PD. To provide a broader theoretical grounding, the literature on phobia is discussed, highlighting the potential of VR as an intervention tool. The review then presents the results of the literature search, summarizing all studies identified through the predefined keywords, and includes a section on interventions in VR applied in PD, although not specifically addressing kinesiophobia but providing evidence of efficacy and feasibility of VR interventions for PD. Finally, insights are proposed to inform the development of E-IVR tailored to reduce kinesiophobia in individuals living with PD.

## 2. The Psychological Basis of Kinesiophobia

In 2011, Knapik et al. contributed significantly to the research into kinesiophobia by broadening the concept and defining it like most phobias [30]. Specifically, they proposed that kinesiophobia encompasses more than just fear and avoidance of physical pain, as assumed when the concept was used originally with chronic and acute pain populations, and suggested it can also stem from a variety of causes, including fear and avoidance of fatigue, exhaustion or social embarrassment [30]. Central to their argument is the idea that kinesiophobia is similar to other specific phobias; it is driven by the perception of threat and maintained by avoidance [31,32,33]. Avoidance behaviors are also described as a part of the kinesiophobia fear-avoidance model, originally developed to explain movement avoidance in chronic musculoskeletal pain populations [34,35]. According to this cognitive-behavioral model, catastrophic interpretations of pain lead to a cycle of fear, avoidance, and hypervigilance, which paradoxically exacerbates the condition and contributes to long-term disability [36,37]. More recently, Slepian et al. [38] expanded the model by incorporating positive psychological constructs such as resilience and self-efficacy. Their findings showed that higher levels of kinesiophobia were associated with poorer three-month outcomes in pain intensity, physical dysfunction, and depressive symptoms, positioning kinesiophobia as a key psychological mediator in their expanded fear-avoidance model. These results reinforce the psychological underpinnings of kinesiophobia and its central role within this framework, where avoidance is conceptualized as a primary coping mechanism in response to perceived threat [30,35,39]. While the specific triggers of kinesiophobia in PD have yet to be clearly established, perceived threat may arise from gait disturbances, fear of falling, freezing episodes or reduced confidence in balance, rather than solely from pain [18,20,23]. In this context, the fear-avoidance mechanism may operate similarly to that described in chronic pain, but it is activated by different sources of threat. However, the specific threats that trigger kinesiophobia in PD remain insufficiently characterized. As the disease progresses, impairments in balance, postural stability and gait tend to worsen, which may increase avoidance tendencies; indeed, higher PD severity has been reported to be associated with higher levels of kinesiophobia [20].

Cognitive Behavioral Therapy (CBT), a well-established treatment for anxiety and phobias, incorporates exposure-based interventions to help individuals develop more adaptive responses to feared stimuli [40,41]. A meta-analysis of the most impactful factors involved in CBT identified exposure to be particularly effective in addressing maladaptive approach-avoidance patterns [42]. Adapting exposure to specific feared cues and avoidance strategies [39,43] is especially critical given that excessive avoidance is the key mechanism maintaining anxiety and phobias. Exposure-based interventions consistently outperform placebo and other psychotherapeutic approaches in treating fear and avoidance behaviors [41,44]. In the context of kinesiophobia, a scoping review of randomized clinical trials that targeted kinesiophobia in people with chronic pain revealed that in vivo exposure to physical activity is the most commonly employed intervention strategy [45], but its efficacy in reducing kinesiophobia has yet to be determined.

Exposure can be conducted in several ways, from using real life situations to VR [43,46]. In vivo exposure involves directly confronting a feared object, situation, or activity in real life, while imaginal exposure requires the patient to vividly imagine the feared situation. Interoceptive exposure focuses on intentionally inducing physical sensations that are harmless but fearful, such as increased heart rate or shortness of breath. The most recent and innovative form, exposure in VR, uses immersive technology to simulate feared situations. This approach is also referred to as in virtuo exposure. In vivo exposure remains the gold standard for treating specific phobias due to its direct approach and proven effectiveness; however, in virtuo is emerging as a promising alternative, offering immersive and controlled environments that help overcome some of the limitations of in vivo methods as detailed below [47].

In vivo exposure is complicated by a number of issues, including its high cost when accompanying patients in various locations or purchasing stimuli are required, the time-consuming nature of the process, and the practical implementation difficulties such as scenarios carried out in public without attracting attention and empathic responses from bystanders. Furthermore, it is possible that patients may be reluctant to engage with such exposure, as they may be unwilling to confront their fear [48]. In order to overcome the aforementioned difficulties, researchers and clinicians are increasingly turning to the use of VR as a tool to facilitate exposure [49,50]. In virtuo exposure enables an ecological, systematic, and gradual approach to stimulus management. By providing a controlled and risk-free environment, VR also enhances the safety of exposure, allowing therapists to focus more on observing and addressing the patient’s avoidance patterns and safety behaviors rather than managing real-world challenges [51]. In general, in virtuo exposure is considered more acceptable than in vivo exposure [48]. Fearful individuals tend to exhibit a higher preference for in virtuo exposure and greater willingness to participate. This could lead to more individuals with phobias receiving treatment. This increased acceptability is supported by participants holding fewer negative beliefs about in virtuo exposure such as perceiving less difficulty tolerating anxiety or distress, and a smaller risk of harm compared to in vivo exposure [52].

It is now well established that despite being aware that the situation is the result of a computer-generated simulation, participants within the context of VR do experience feelings and responses that are comparable to those experienced in real-world scenarios [53,54,55]. The phenomenon is characterized by a sense of presence, which is contingent on the immersive quality of the virtual experience. Immersion, in this context, refers to the technology’s ability to deliver coherent multisensory inputs while isolating the user from the real-world environment, such as a laboratory setting. This immersive engagement fosters two fundamental illusions: the illusion of place, engendering the impression of actually being within the simulated environment; and the illusion of plausibility, engendering the sense that the virtual events unfolding are occurring in reality [56]. Collectively, these perceptual illusions contribute to the powerful feeling of presence in virtual contexts which is a significant component in the treatment of phobias.

Given its strong empirical support in the treatment of other phobias and its theoretical alignment with the fear-avoidance model, E-IVR, as part of CBT, appears particularly relevant for the treatment of kinesiophobia in people with PD. While treatments for kinesiophobia have been more extensively documented in other populations [45], no intervention to date has specifically targeted kinesiophobia in PD, further highlighting the clinical need for E-IVR approaches.

## 3. Virtual Reality-Based Exposure Intervention as a Treatment of Kinesiophobia

The potential of VR to target the fear-avoidance scheme exhibited by individuals with kinesiophobia suggests its potential as a therapeutic modality for managing the fear of movement [57]. A number of studies have been conducted examining the utilization of VR as a tool for reducing chronic pain and targeting kinesiophobia as a main or secondary outcome (Table 1). For instance Wang et al. [58] conducted a meta-analysis with no restriction on the type of disease, which demonstrated a significant reduction in kinesiophobia following E-IVR, as measured as a secondary outcome by the Tampa Scale for Kinesiophobia (TSK) and the Fear-Avoidance Beliefs Questionnaire (FABQ), particularly among patients with chronic low back pain (effect size standard mean difference of −0.59 (95% CI [−0.95, −0.22], *p* = 0.002)). Notably, this study found that semi-immersive platforms such as Nintendo Wii and Xbox Kinect were more effective than fully immersive systems, suggesting that ease of use of the VR system, the sense of presence, or perceived realism, may modulate outcomes. A reduction of kinesiophobia in chronic low back pain patients has also been reported by another meta-analysis conducted by Li et al. [59] which reported clinically meaningful immediate improvements in pain-related fear following VR-based exercise training. However, these effects were not sustained after the intervention (3 to 6 months), and the overall quality of evidence was deemed low. While these findings are promising, some nuances deserve attention. It is worth noting that the results from the study presented below are based on fear of movement assessed with the Falls Efficacy Scale which is neither a direct nor validated measure of kinesiophobia. Indeed, a systematic review focusing on older adults having a fear of movement, with or without pain, found that while VR training improved physical mobility, its effects on fear itself were inconsistent and statistically non-significant [60]. Two other studies, which were not randomized controlled trials and therefore not included in the aforementioned meta-analyses, add further complexity. Fowler et al. [61] assessed the feasibility of an E-IVR for veterans with chronic pain. While the Pain Outcomes Questionnaire (POQ-VA) showed little evidence for kinesiophobia reduction, the Fear of Daily Activities Questionnaire (FDAQ) indicated that 38% of veterans exceeded the minimum clinically important difference, with a small effect size improvement observed. The authors suggested that activity-specific measures like the FDAQ may better capture changes in kinesiophobia in this population than general scales such as POQ-VA. Similarly, Chen et al. [62] studied individuals with chronic neck pain performing neck exercises in a VR environment where visual feedback was intentionally manipulated to alter perceived movement. Participants increased their neck rotation when the visual display showed less movement than they were actually performing, suggesting that VR may promote movement even in the presence of fear. However, these behavioral improvements were not directly associated with changes in self-reported kinesiophobia scores.

Overall, current evidence suggests that VR interventions consistently improve physical movement across a range of populations. However, findings related to kinesiophobia itself are inconsistent. This inconsistency may be attributed to the fact that most interventions are primarily designed as rehabilitative interventions to promote movement and distract from pain, with kinesiophobia assessed only as a secondary outcome. As described above, physical rehabilitation interventions have a different focus: they aim to progressively reduce avoidance, promote safety learning, and build self-efficacy to engage in action despite feared outcomes [39,64]. Moreover, previously mentioned findings have emphasized the selection of the measurement tool, often other than the TSK, the design of the E-IVR scenario, the type of VR technology employed, and its integration with physical activity are critical factors influencing intervention effectiveness [59]. Furthermore, as highlighted by the studies discussed above, individuals with chronic pain remain the primary population in which E-IVRs have been used to assess or reduce kinesiophobia. Although these findings offer valuable theoretical insights, the fear-avoidance mechanisms that characterize kinesiophobia are likely generalizable to PD, given that kinesiophobia reflects a form of phobic fear comparable to that studied in other populations. What differs in PD are the potentially specific triggers that activate the fear–avoidance cycle, such as balance impairments, freezing of gait, or increased risk of falls rather than fear of pain. Therefore, E-IVR may remain highly relevant for PD, especially considering the ecological, controlled and risk-free environment advantages of in virtuo, but must be tailored to address these disease-specific sources of perceived threat. To date, no study has specifically employed E-IVR to target kinesiophobia in individuals with PD, despite its acknowledged clinical relevance in this population.

To further illustrate the potential role of exposure, such as E-IVR, for addressing kinesiophobia in PD, we propose a hypothetical model (Figure 1) that adapts well-established anxiety-maintenance processes from other phobias [65] to the context of kinesiophobia in PD. Although the exact psychological mechanisms of kinesiophobia have not yet been formally specified in PD, the model outlines how PD-specific triggers may elicit a perceived threat that could induce kinesiophobia. When facing fearful situations, a typical reaction is to avoid the stimuli and context that triggered the fear. However, this behavior is setting a cognitive trap, where avoidance behaviors prevent people to learn they are overestimating the consequences, likelihood or imminence of what they are afraid of, or that they may be able to effectively cope with it [66]. This leads to a vicious circle, ultimately leading to reduced physical activity and physical deconditioning. Exposure-based intervention, for example, in the form of E-IVR, could be used to help individuals with PD engage with the perceived threat, thereby preventing avoidance, reshaping threat appraisals, and progressively breaking the vicious cycle to reduce kinesiophobia.

## 4. Feasibility and Efficacy of VR Interventions in Individuals Living with PD

Although the interventions described hereafter do not specifically address kinesiophobia, the use of VR in PD highlights promising and inspiring directions for developing E-IVR targeting kinesiophobia. Two VR interventions have already been employed to target psychological well-being of individuals with PD. A study evaluating 12 sessions of E-IVR found significant improvements in well-being among individuals with PD [67]. The intervention was designed to offer a controlled and interactive environment for skill practice, with the aim of enhancing general psychological wellbeing and promoting emotional resilience. Kinesiophobia was not specifically targeted in that study, but the intervention group showed a reduction in anxiety and depression scores and an improvement in quality of life compared to the control group. Another study also investigated the effect of an at-home intervention consisting of watching a self-modelling video of personalized exercises using VR for individuals experiencing freezing of gait [68]. While overall reductions in anxiety were modest, the intervention produced notable improvements in participants with high baseline anxiety with four out of five showing marked decreases on the Parkinson Anxiety Scale. Qualitative feedback also indicated increased confidence and reduced anxiety in situations that are typically associated with freezing episodes. Because anxiety and low safe confidence are core components of movement-related fear in PD, often contributing to avoidance of situations that may trigger freezing, these findings plausibly reflect a reduction in fear of movement, even though kinesiophobia was not directly measured. These evidence suggest that E-IVR may be effective in reducing anxiety and, in turn, improving motor outcomes in people living with PD. These findings on anxiety are promising for the use of E-IVR in reducing kinesiophobia. In addition, several studies, summarized in three systematic reviews and meta-analyses, have explored the use of VR in the field of neurorehabilitation. Although they did not specifically address kinesiophobia, their results confirm the benefits of VR for balance, and to a lesser extent, for gait and motor function in individuals with PD [63,69,70]. These outcomes are highly relevant to kinesiophobia, as improvements in balance, gait stability, and perceived motor control are known to reduce fear of falling and enhance confidence in movement. Thus, even without directly assessing kinesiophobia, these findings further support the potential relevance of E-IVR for PD populations that may experience movement-related fear or avoidance.

Beyond efficacy, the feasibility and acceptability of VR interventions designed for people living with PD have also been explored. For example, Brandín-De la Cruz et al. [71] reported a 97% adherence rate to a treadmill-based VR program, while Goh et al. [68] observed a 90% retention rate and high adherence to prescribed video viewing and physical practice in this population. Importantly, VR interventions were generally found to be safe for people living with PD. For instance, a study on somato-cognitive coordination intervention [72] and another one on VR-based balance training [73] reported no significant adverse events. Studies have reported high levels of satisfaction and usability with VR interventions aimed at promoting physical activity among people living with PD, indicating that patients find these technologies acceptable and beneficial. Specifically, in one study, the use of a VR headset paired with a boxing exergame was found to be safe, well-tolerated, and positively received, with high usability and satisfaction reported over 60 sessions [74]. Another study used a VR headset with serious games targeting upper limb rehabilitation and found improvements in strength, coordination, and dexterity, along with full adherence and no adverse effects [75]. These findings collectively support the potential use of VR interventions aimed at reducing kinesiophobia in people living with PD, as they are reported to be safe, well-tolerated, and effective in improving motor symptoms, reducing non-motor symptoms such as anxiety, and promoting overall engagement in physical activity.

## 5. Insight into Designing Exposure-Based Interventions in VR Targeting Kinesiophobia in PD

An important component of VR experience is the design of the virtual environment, especially in the context of clinical exposure-based interventions for treating phobias such as kinesiophobia. In their review, Thangavelu et al. [29] proposed five key design principles to guide the development of anxiety-focused VR protocols: (1) identifying and addressing PD-specific sources of anxiety, (2) simplifying patient input and interface design when immersed in VR, (3) prioritizing safety and gradual immersion, (4) incorporating real-time visual, auditory, and bio feedback, and (5) ensuring hardware comfort and visual accessibility.

Building on these findings, our review of the existing literature on VR interventions targeting PD-specific sources of anxiety or physical exercise, without addressing kinesiophobia, for rehabilitation leads to the identification of two main E-IVR approaches used with people living with PD. The first approach, which may be used to reduce kinesiophobia in people living with PD, involves positive contact with fear through distraction techniques and cognitive restructuring, which has also shown promise. For example, Zhang et al. [76] used a butterfly-catching game in which patients stood on virtual grass and reached for moving butterflies, fostering motor engagement in a non-threatening context. The second approach involves gradual or well targeted exposure to the feared stimuli (no distraction used) which may be applied to people living with PD to alleviate kinesiophobia. This approach is well supported and has demonstrated significant reductions in anxiety and avoidance behaviors in the context of specific phobias, therefore it is widely used [40,41,43,51]. For instance, Landers and Nilsson [77] created VR environments for individuals with PD simulated common fear of falls triggers such as crowds, stairs, doorways, and narrow corridors. Using a head-mounted display, participants completed functional tasks like walking, turning, and navigating obstacles, while postural, gait, and neurophysiological responses were assessed. Similarly, Pérez-Sanpablo et al. [78] developed a VR-based treadmill featuring three immersive virtual environments (i.e., indoor hallway, a beach, and a forest) with infinite hallway to allow continuous walking for PD gait rehabilitation. A key design feature was real-time visual feedback using virtual footprints to guide gait correction.

Rather than being used separately, these two approaches could hold potential for kinesiophobia treatment in PD by being combined within a single E-IVR. This is well illustrated in a study by Finley et al. [79] evaluating the usability of “Wordplay VR”, a VR-based mobility training game for people with PD, designed to enhance the motivation to engage in physical activity, while also promoting cognitive engagement within a fully immersive 3D environment. Participants solved word puzzles by collecting floating virtual letters while walking, turning, reaching, and navigating virtual obstacles such as barriers and stepping challenges which were adjustable. By blending elements of graded exposure (virtual obstacles) with cognitive distraction (word puzzles) and engagement, this study demonstrated how a dual-approach E-IVR can address both psychological and functional aspects of fear-related behaviors in PD.

Building on these insights, several directions for future research emerge. First, although a kinesiophobia scale specifically developed for PD already exists (TSK-PD) [22], further validation, cross-cultural adaptation, and evaluation of its sensitivity to change in the context of VR-based interventions are needed. Second, there is a clear need for the development and testing of pilot E-IVR protocols explicitly designed to target kinesiophobia in PD, integrating disease-specific triggers such as gait disturbances, fear of falling, or freezing episodes. Third, future trials should evaluate kinesiophobia on equal footing with fear-of-falling and mobility-related outcomes, as it provides additional insight into the cognitive behavioral mechanisms that drive avoidance in PD. Finally, long-term follow-up assessments will be essential to determine whether reductions in kinesiophobia are sustained over time and translate into meaningful improvements in participation, confidence, and daily functioning.

## 6. Conclusions

Kinesiophobia is common in PD and may stem from a gradual loss of movement control and social judgment. Its role in the fear-avoidance model highlights the need for interventions addressing the psychological mechanisms underlying avoidance behaviors. Exposure-based interventions, particularly in VR, have shown promise in reducing kinesiophobia in chronic pain populations. These findings suggest that VR may support movements by offering safe and engaging environments. However, to ensure the effectiveness of E-IVR for reducing kinesiophobia, it is essential to adopt reliable and consistent methods to specifically measure kinesiophobia. Indeed, a wide range of assessment tools was identified in studies applying VR interventions. This methodological variability complicates the interpretation of outcomes and hinders the ability to determine the extent to which E-IVR effectively reduces kinesiophobia across diverse populations. Although no study to date has directly targeted kinesiophobia in PD using VR, existing VR interventions have demonstrated improvements in anxiety and motor symptoms. We propose that combining graded exposure with cognitive and motor engagement may offer an effective way to reduce both psychological barriers, such as kinesiophobia, and functional barriers to physical activity in people living with PD. Considering that VR appears to be a feasible, acceptable, and safe therapeutic tool in this population, future research should build on existing VR design principles in PD to develop targeted E-IVR.

## Figures and Tables

**Figure 1 jcm-14-08837-f001:**
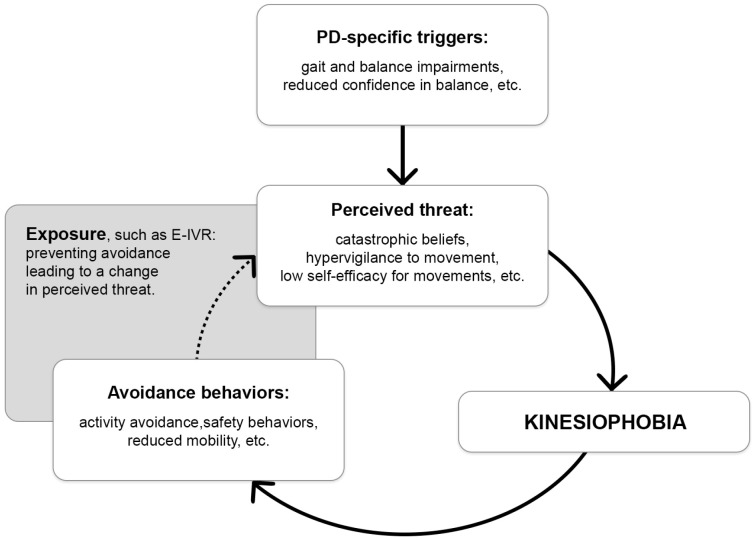
Hypothetical kinesiophobia maintaining model in Parkinson’s disease and proposed exposure-based intervention mechanism of action. The grey box illustrates where the exposure-based intervention would act, and the dotted arrow indicates where an intervention such as E-IVR could disrupt the vicious cycle by supporting individuals with PD in engaging with the perceived threat. In turn, this may limit avoidance, adjust threat appraisals, and progressively reduce the processes that sustain kinesiophobia. Parkinson’s disease (PD); Exposure-based intervention in virtual reality (E-IVR).

**Table 1 jcm-14-08837-t001:** Summary of exposure-based intervention in virtual reality for kinesiophobia studies included in the narrative review.

Study	Population	Design and Sample Size (n)	Immersion Type and Type of Intervention	Measures of Kinesiophobia	Main Outcomes	Limitations
Li et al. (2024) [59]	Adults with chronic low back pain	Systematic review + meta-analysis RCT 20 RCTs included (n ≈ 1059)	Mixed: non-immersive (Nintendo Wii, Kinect) and fully immersive Various VR exercise or rehabilitation programs	TSK, FABQ	VR improved pain and disability; small to moderate reduction in kinesiophobia short-term; no long-term effect; high heterogeneity across studies.	Low-quality evidence; small samples; inconsistent follow-up; kinesiophobia not consistently measured across studies
Wang et al. (2019) [63]	Adults with musculoskeletal pain (no restriction on the type of disease)	Meta-analysis of RCT: 11 RCT (n ≈ 645)	Mixed: non-immersive (Nintendo Wii, Kinect) and fully immersive VR-based movement therapy, VR exercise, VR distraction	TSK, FABQ	VR significantly reduced kinesiophobia; Results inconsistent across studies; Non-immersive technology more effective than fully immersive	Heterogeneity of VR protocols; kinesiophobia not primary target; inconsistent measurement tools
Percy et al. (2023) [60]	Older adults with fear of falling & mobility impairments	Systematic review + meta-analysis of RCT: 7 RCT (n ≈ 297)	Mixed: non-immersive (Nintendo Wii, Kinect) and one fully immersive VR-based exercise or balance training	FES	VR improved mobility but did not significantly reduce fear of movement	Use of non-specific fear-of-movement measures; inconsistent exposure intensity across studies.
Fowler et al. (2019) [61]	Veterans with chronic pain	Single-arm feasibility study, n = 16	Fully immersive graded VR exposure therapy	POQ, FDAQ	38% reached minimum clinically important difference on FDAQ; small improvements in movement fear POQ	No control group; small sample; hierarchy design issues
Chen et al. (2017) [62]	Adults with chronic neck pain	Two single-session experimental studies (within-subject), asymptomatic vs. chronic neck pain. Chronic pain n = 9 Asymptomatic n = 10	Semi-immersive VR with manipulated visual feedback during neck rotation. Visuomotor perturbation VR task (sensorimotor manipulation).	TSK	Increased movement under altered visual feedback; kinesophobia assessed but not evaluated for change (no pre–post measurement).	Kinesiophobia not primary focus; lacks standardized measures; small sample

Note. Randomized Clinical Trials (RCT); Tampa Scale for Kinesiophobia (TSK); Fear-Avoidance Beliefs Questionnaire (FABQ); Pain Outcomes Questionnaire (POQ); Falls Efficacy Scale (FES); Fear of Daily Activities Questionnaire (FDAQ). FABQ, FES, POQ, and FDAQ are not validated to measure kinesiophobia.

## Data Availability

Not applicable as this is a narrative review.

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
