# Peer review of "Exposure-Based Intervention in Virtual Reality to Address Kinesiophobia in Parkinson’s Disease: A Narrative Review"

_jcm, 2025, doi:10.3390/jcm14248837_

Round 1

Reviewer 1 Report

Comments and Suggestions for Authors

This manuscript addresses an important and underexplored topic: the potential use of exposure-based virtual reality (VR) interventions to reduce kinesiophobia in Parkinson’s disease (PD). The narrative review is relevant and timely, especially given the growing role of digital therapeutics in movement disorders. The paper is well organized, generally readable, and supported by an extensive literature base. However, The review currently reads more like a broad descriptive overview rather than a deep critical synthesis. The rationale for several methodological decisions is insufficiently justified. So, it needs some clarifications that would help improve the quality before publication.

1- The manuscript repeatedly suggests that kinesiophobia in PD is “understudied” and may differ from musculoskeletal pain populations. This is correct, but the paper does not fully articulate:

  • How kinesiophobia manifests differently in PD

  • Which psychological mechanisms dominate (fear of falling vs fear of pain)

  • How kinesiophobia interacts with disease stage, freezing, festination, and balance impairments

This is essential to justify why VR exposure would be theoretically effective in PD. I suggest to expand the introduction section to more clearly differentiate PD-specific kinesiophobia from general fear of movement models.

2- Provide a concise table/figure summarizing the included VR–kinesiophobia studies, highlighting design, outcomes, measures used, immersion type, and limitations.

3- Adding a conceptual model figure illustrating the proposed E-IVR mechanism in PD would improve clarity.

Thanks for your great work and wish you all the best.

Author Response

REVIEWER #1

1- The manuscript repeatedly suggests that kinesiophobia in PD is “understudied” and may differ from musculoskeletal pain populations. This is correct, but the paper does not fully articulate:

  • How kinesiophobia manifests differently in PD

Response & Action: Thank you for this insightful comment. We agree that the manuscript did not sufficiently articulate how kinesiophobia may manifest differently in Parkinson’s Disease (PD). We have now added text to clarify that while the original fear-avoidance model was developed in the context of musculoskeletal pain, the triggers of perceived threat in PD are distinct. We also added more information about how kinesiophobia and its manifestations are different in PD. These changes can be found at lines 87-88, 93-96, 179-187 and 243 to 246.

  • Which psychological mechanisms dominate (fear of falling vs fear of pain)

Action: The distinction between kinesiophobia, fear of falling and fear of pain has been made clearer, and details about the hypothesized mechanisms of kinesiophobia in PD have been added at lines 179-187.

  • How kinesiophobia interacts with disease stage, freezing, festination, and balance impairments

Action: Known relationships between kinesiophobia and other factors such as disease stage and balance have been added in the Introduction section at lines 93 to 96. The hypothesized role of disease stage, gait impairments and balance impairments in the psychological basis of kinesiophobia has been added at lines 179 to 187 and in Figure 1 at page 9.

This is essential to justify why VR exposure would be theoretically effective in PD. I suggest to expand the introduction section to more clearly differentiate PD-specific kinesiophobia from general fear of movement models.

Response: Thank you for this suggestion. The advantages of E-IVR are outlined at lines 206 to 228, and the rationale for using it specifically for kinesiophobia in Parkinson’s was added at lines 306 to 313. As detailed above, changes regarding your other comments have been made accordingly in the Introduction section at lines 93 to 96, but also in the sections ‘2. The psychological basis of kinesiophobia’ at lines 179 to 187 and 243 to 246 and ‘3. Virtual reality-based exposure intervention as a treatment of kinesiphobia’, as well as in Figure 1 at lines 306 to 313 and 317 to 331.

2- Provide a concise table/figure summarizing the included VR–kinesiophobia studies, highlighting design, outcomes, measures used, immersion type, and limitations.

Response & Action: We would like to thank the reviewer for bringing to our attention the added value of including a table summarizing the VR-related kinesiophobia literature. Table 1 is now available at pages 7 and 8 and provides a concise summary of the studies involving E-IVR for kinesiophobia, highlighting their designs, immersion characteristics, measurement tools, outcomes, and main limitations.

3- Adding a conceptual model figure illustrating the proposed E-IVR mechanism in PD would improve clarity.

Response & Action: Thank you for this helpful suggestion. We agree that a conceptual model would improve the clarity of our discussed mechanism and E-IVR role. Figure 1 is now available at page 9, and provides a hypothetical kinesiophobia maintaining model in PD and proposed exposure-based intervention mechanism of action.

Thanks for your great work and wish you all the best.

Response: Thank you for your insightful comments.

Reviewer 2 Report

Comments and Suggestions for Authors

Dear authors , in summary, the manuscript provides a comprehensive and insightful review on a niche but impactful topic. The follow suggestions aim to further strengthen its clarity and completeness. The authors should address the methodological clarifications, expand on a couple of conceptual points (relationship of kinesiophobia to known PD fears, and future directions), and correct minor errors. These revisions will polish the work.

Please consider the following specific revisions:

1. Structure, Organization, and Presentation of Evidence

1.1. Add a summary table of key studies:

  • Include at least one table summarizing:

    • (a) VR interventions targeting kinesiophobia or fear of movement in chronic pain/neurological or musculoskeletal conditions, and

    • (b) VR interventions in PD.

  • For each study, indicate: population, condition, sample size, type of VR intervention, level of immersion, primary/secondary outcomes (especially kinesiophobia or related fear measures), and main findings.

1.2. Add a brief recap at the end of the section summarizing the literature search results (before the design-insights section):

  • Explicitly state how evidence from other populations (e.g., chronic pain) translates to PD and where the main knowledge gaps in PD remain.

2. Methodology and Search Strategy

2.1. Provide more detailed information on the literature search:

  • List the databases searched (e.g., PubMed, Scopus, Web of Science, PsycINFO, etc.).

  • Indicate the main search date range (e.g., from year X to April 2025).

  • Specify key inclusion and exclusion criteria (e.g., population type, language, study design, outcome measures).

2.2. Consider including a simple flow description of study selection:

  • Provide at least the number of records identified, screened, excluded, and finally included.

  • If possible, add a brief PRISMA-style flow diagram as a figure or in supplementary material to strengthen transparency.

3. Conceptual Clarifications (Kinesiophobia vs Related Constructs)

3.1. Clarify the relationship between kinesiophobia and fear of falling in PD:

  • Explain explicitly whether kinesiophobia in PD is mainly driven by fear of falling or whether it encompasses additional concerns (e.g., fear of worsening symptoms, social embarrassment, fatigue).

  • Clarify conceptual boundaries and overlap between these constructs in PD.

3.2. Briefly discuss existing non‑VR strategies that may touch on fear of movement/fear of falling in PD:

  • Mention any known approaches (e.g., balance training with anxiety management, CBT-based approaches, education, gait training with confidence-building components), if they exist.

  • If no interventions specifically address kinesiophobia or fear of movement in PD, state this clearly to underline the novelty and clinical need for VR exposure-based approaches.

3.3. Add a short comment on terminology:

  • Acknowledge that the term “kinesiophobia” originates from chronic pain research and that in PD-related literature clinicians may use terms such as “fear of falling,” “fear of movement,” or “movement avoidance.”

  • Clarify how these terms are used in the present manuscript and how they map onto the broader concept of kinesiophobia.

4. Use of Evidence and Future Directions

4.1. Make more explicit how VR studies in PD relate to movement-related fear:

  • In the section on VR in PD, explicitly highlight which outcomes are most relevant to kinesiophobia (e.g., anxiety reduction, increased confidence, greater willingness to move).

  • For example, for the Goh et al. [63] study, add one or two sentences explaining that reduced anxiety and improved confidence in high-anxiety participants plausibly relate to reduced movement-related fear, even if kinesiophobia was not directly measured.

4.2. Expand the future research directions in the conclusion:

  • Propose concrete next steps such as:

    • Validation or adaptation of kinesiophobia scales specifically for PD.

    • Design and testing of pilot exposure-based VR protocols directly targeting kinesiophobia in PD.

    • Recommendations on appropriate primary and secondary outcome measures (e.g., specific kinesiophobia scales, fear of falling scales, objective activity measures).

    • Consideration of long‑term follow-up to assess durability of kinesiophobia reductions.

5. References and Formatting

5.1. Ensure complete bibliographic information for all references:

  • For references that currently appear mainly as “Available from:” with a URL, add:

    • Journal name,

    • Year,

    • Volume and issue,

    • Page range or article number,

    • DOI where available.

5.2. Check consistency with JCM reference style:

  • Harmonize punctuation, abbreviations of journal titles, use of italics, and ordering of elements in each reference.

6. Language, Style, and Minor Corrections

6.1. Correct typographical errors:

  • Correct “Knesiophobia” to “Kinesiophobia” in the title and anywhere else it appears.

  • Scan the manuscript for additional minor typos or spacing issues (e.g., unnecessary spaces before punctuation, inconsistent hyphenation).

6.2. Slightly shorten or split overly long sentences:

  • Review sections with particularly long, complex sentences (e.g., in the introduction and theoretical sections) and break them into shorter units where appropriate, without changing the scientific meaning.

6.3. Ensure consistent use of abbreviations and terminology:

  • Check that abbreviations such as PD, VR, E‑IVR, CBT, etc., are defined at first use and consistently applied thereafter.

  • Use terms like “kinesiophobia,” “fear of movement,” “fear of falling” consistently according to the clarified definitions.

6.4. Re‑check for any awkward phrasing possibly introduced during AI‑assisted proofreading:

  • Carefully reread the manuscript to adjust any phrasing that sounds slightly unnatural or repetitive and align it with standard academic style.

Author Response

REVIEWER #2

  1. Structure, Organization, and Presentation of Evidence

1.1. Add a summary table of key studies:

Include at least one table summarizing:

(a) VR interventions targeting kinesiophobia or fear of movement in chronic pain/neurological or musculoskeletal conditions, and

(b) VR interventions in PD.

For each study, indicate: population, condition, sample size, type of VR intervention, level of immersion, primary/secondary outcomes (especially kinesiophobia or related fear measures), and main findings.

Response & Action: Thank you for this helpful comment. In response, we have added a comprehensive summary table (Table 1) that can be found at page 7 and 8 that provides, for each study included in our narrative review, the population, condition, sample size, type of VR intervention, level of immersion, the kinesiophobia or related fear measures used, as well as the main findings and studies limitations. Regarding VR interventions conducted specifically in PD, we appreciate the suggestion. However, our article does not aim to systematically review all VR interventions in PD studied (i.e., those not linked to kinesiophobia). The VR interventions in PD studies were mentioned only to illustrate current conceptual or technological directions and were not part of the evidence base of the narrative review itself. For this reason, and to avoid misrepresenting the scope of our review, we did not include PD-specific VR interventions in the summary table. We clarified this distinction in the manuscript to ensure consistency and transparency.

1.2. Add a brief recap at the end of the section summarizing the literature search results (before the design-insights section): Explicitly state how evidence from other populations (e.g., chronic pain) translates to PD and where the main knowledge gaps in PD remain.

Action: Thank you for this suggestion. A few sentences highlighting how the evidence from other populations can generalize to Parkinson’s, but also how kinesiophobia triggers in PD can be unique, were added to summarize the literature detailed earlier on, and this can be found at lines 305 to 313.

  1. Methodology and Search Strategy

2.1. Provide more detailed information on the literature search:

- List the databases searched (e.g., PubMed, Scopus, Web of Science, PsycINFO, etc.).

- Indicate the main search date range (e.g., December 2024 to April 2025).

- Specify key inclusion and exclusion criteria (e.g., population type, language, study design, outcome measures).

Action: Information was added to detail the process of our literature search. Databases used and the search dates were added at lines 125 to 127. Our search strategy is now further detailed at lines 131 to 139.

2.2. Consider including a simple flow description of study selection:

Provide at least the number of records identified, screened, excluded, and finally included.

Action: These details, along with more information on our search strategy is now available at lines 140 to 149.  

If possible, add a brief PRISMA-style flow diagram as a figure or in supplementary material to strengthen transparency.

Response: We fully agree that transparency in the review process is important. However, because the present work is a narrative review rather than a systematic review, a formal PRISMA flow diagram does not apply to our methodology. Narrative reviews do not follow predefined systematic search, screening, and selection procedures required by PRISMA, and applying this framework could unintentionally misrepresent the scope and aims of the paper. That being said, to enhance transparency, we have, as described above, clarified our search strategy, databases consulted, key search combinations, and full-text screening steps at lines 125 to 127, 131 to 139, and 140 to 149. We believe this level of details is appropriate for a narrative review while accurately reflecting the exploratory nature of the field.

  1. Conceptual Clarifications (Kinesiophobia vs Related Constructs)

3.1. Clarify the relationship between kinesiophobia and fear of falling in PD:

- Explain explicitly whether kinesiophobia in PD is mainly driven by fear of falling or whether it encompasses additional concerns (e.g., fear of worsening symptoms, social embarrassment, fatigue).

- Clarify conceptual boundaries and overlap between these constructs in PD.

Response & Action: Thank you for this comment. We agree that it is important to clarify whether kinesiophobia in PD is primarily driven by fear of falling or whether it reflects a broader set of concerns. We have now added at lines 371 to 375 and 381 to 385 an explicit statement in the manuscript indicating that, while fear of falling is a major driver of kinesiophobia in PD, additional disease-specific triggers, such as fear of worsening motor symptoms, freezing of gait, balance loss, fatigue, or social embarrassment, may also activate the fear–avoidance cycle. We also clarified the conceptual boundaries and areas of overlap between these constructs in PD, building on the additions already made in response to previous comments. This should help distinguish the specific contributions of fear of falling from the broader psychological mechanisms involved in kinesiophobia.

3.2. Briefly discuss existing non‑VR strategies that may touch on fear of movement/fear of falling in PD:

- Mention any known approaches (e.g., balance training with anxiety management, CBT-based approaches, education, gait training with confidence-building components), if they exist.

- If no interventions specifically address kinesiophobia or fear of movement in PD, state this clearly to underline the novelty and clinical need for VR exposure-based approaches.

Response & Action: We have now briefly outlined existing non-VR strategies that may indirectly influence fear of movement or fear of falling in PD, such as balance training combined with confidence-building, physiotherapy integrating cognitive-behavioral elements, fall-prevention programs, and educational approaches at lines 381 to 385. At lines 243 to 246, we also now highlight that, although treatments for kinesiophobia have been tested in several clinical populations, no intervention has yet targeted kinesiophobia directly in PD. Moreover, no exposure-based framework has been applied to PD despite its theoretical relevance. We have clarified this point in the manuscript at lines 381 to 385 to reinforce the novelty and clinical need for VR exposure-based approaches in this population.

3.3. Add a short comment on terminology:

- Acknowledge that the term “kinesiophobia” originates from chronic pain research and that in PD-related literature clinicians may use terms such as “fear of falling,” “fear of movement,” or “movement avoidance.”

- Clarify how these terms are used in the present manuscript and how they map onto the broader concept of kinesiophobia.

Response & Action: Thank you for this suggestion. We would like to first point out that the origine of ‘’kinesiophobia’’ is already mentioned at lines 85 to 90 of the manuscript. Secondly, in this article, we describe and discuss kinesiophobia; this is thus the term we use throughout the manuscript. However, we use the term ‘fear of movement’ or ‘movement avoidance’ when discussing results from articles that used these terms specifically instead of kinesiophobia. To clarify these nuances, at lines 99 and 105, we now specify how ‘fear of movement’ or ‘movement avoidance’ can be used interchangeably in the literature by clinicians and researchers, and that in our manuscript we use the term ‘kinesiophobia’ unless other terms were employed in the discussed articles. Furthermore, it has been clarified throughout the manuscript how fear of falling is a different, yet related, concept than kinesiophobia.

  1. Use of Evidence and Future Directions

4.1. Make more explicit how VR studies in PD relate to movement-related fear:

- In the section on VR in PD, explicitly highlight which outcomes are most relevant to kinesiophobia (e.g., anxiety reduction, increased confidence, greater willingness to move).

- For example, for the Goh et al. [63] study, add one or two sentences explaining that reduced anxiety and improved confidence in high-anxiety participants plausibly relate to reduced movement-related fear, even if kinesiophobia was not directly measured.

Action: This explicit clarification was added at lines 371-375 and 381-385

4.2. Expand the future research directions in the conclusion:

Propose concrete next steps such as:

- Validation or adaptation of kinesiophobia scales specifically for PD.

- Design and testing of pilot exposure-based VR protocols directly targeting kinesiophobia in PD.

- Recommendations on appropriate primary and secondary outcome measures (e.g., specific kinesiophobia scales, fear of falling scales, objective activity measures).

- Consideration of long‑term follow-up to assess durability of kinesiophobia reductions.

Action: We would like to thank the reviewer for this suggestion. We have now added a full paragraph discussing future research directions at lines 447-458, which includes the need to develop and test E-IVR protocols, and to use validated measurements of kinesiophobia.

  1. References and Formatting

5.1. Ensure complete bibliographic information for all reference

For references that currently appear mainly as “Available from:” with a URL, add:

Journal name, Year, Volume and issue, Page range or article number, DOI where available.

Action: We would like to thank the reviewer for pointing this out. All the references have been revised and now align with journal formatting requirements.

5.2. Check consistency with JCM reference style:

Harmonize punctuation, abbreviations of journal titles, use of italics, and ordering of elements in each reference.

Action: We would like to thank the reviewer for pointing this out. All the references have been revised and now align with journal formatting requirements.

  1. Language, Style, and Minor Corrections

6.1. Correct typographical errors:

Correct “Knesiophobia” to “Kinesiophobia” in the title and anywhere else it appears. Scan the manuscript for additional minor typos or spacing issues (e.g., unnecessary spaces before punctuation, inconsistent hyphenation).

Response: We would like to thank the reviewer for a careful analysis of our manuscript. After careful re-lecture of our manuscript, we did not notice any ‘knesiophobia’ misspelling. The only place where ‘kinesiophobia’ is spelled differently is in the reference 17. This article from Kori, Miller and Todd (1990) is the seminal paper on kinesiophobia. At that time, the spelling ‘kinisophobia’ was used. We kept that spelling in the article reference as it corresponds to the appropriate title for the article. We will be happy to correct any other misspelled ‘kinesiophobia’ you may find in our manuscript.

6.2. Slightly shorten or split overly long sentences:

Review sections with particularly long, complex sentences (e.g., in the introduction and theoretical sections) and break them into shorter units where appropriate, without changing the scientific meaning.

Response & Action: Thank you for noticing these long and complex sentences. A careful re-lecture of our manuscript enabled us to break them into shorter and more simple sentences. Such examples can be found at lines 44-49 and 62-66.

6.3. Ensure consistent use of abbreviations and terminology:

Check that abbreviations such as PD, VR, E‑IVR, CBT, etc., are defined at first use and consistently applied thereafter.

Response: We would like to thank the reviewer for a careful analysis of our manuscript. After careful re-lecture of our manuscript, we notice abbreviations that were not used correctly and consistently throughout the text and they have been corrected. Please see lines 21, 27, 124 and 125. At line 499, E-IVR, TSK, FABQ, POQ, FES, and FDAQ were added in the Abbreviation list, however, the abbreviation was already defined in the text. We will be happy to correct any misused abbreviation you may find in our manuscript.

Use terms like “kinesiophobia,” “fear of movement,” “fear of falling” consistently according to the clarified definitions.

Response: A careful re-lecture of our manuscript ensured that all the terms were used appropriately, and we clarified our terminology in lines 99 to 102.

6.4. Re‑check for any awkward phrasing possibly introduced during AI‑assisted proofreading:

Carefully reread the manuscript to adjust any phrasing that sounds slightly unnatural or repetitive and align it with standard academic style.

Response: A careful re-lecture of our manuscript ensured a natural phrasing.
